# The Descemet Membrane Endothelial Keratoplasty (DMEK) “Wave Maneuver”

**DOI:** 10.3390/jcm11185260

**Published:** 2022-09-06

**Authors:** Itay Lavy, Nir Erdinest, Ayala Katzir, Naomi London, Eleanor Ngwe Nche, David Smadja

**Affiliations:** 1Department of Ophthalmology, Hadassah-Hebrew University Medical Center, Faculty of Medicine, Hebrew University of Jerusalem, Jerusalem 91905, Israel; 2Private Practice, Jerusalem 9422805, Israel

**Keywords:** corneal transplantation, Descemet membrane endothelial keratoplasty, DMEK, graft unfolding, Descemet membrane

## Abstract

A novel technique for Descemet membrane endothelial keratoplasty (DMEK) graft handling and centration without the endothelium touching the posterior part of the anterior chamber (AC), is presented here. It is particularly suitable for vitrectomized eyes, deep AC, and AC intraocular lenses (ACIOLs), potentially reducing surgery time and endothelial cell loss during surgery. This retrospective interventional case series includes 27 eyes with complex ocular pathology. All utilized a “Wave maneuver” to center an early elevated graft without completing graft centration on the bottom of the AC. Successful graft attachment and centration were evaluated intra and post-operatively. Best-corrected visual acuity (BCVA), central corneal thickness (CCT), and donor endothelial cell density (ECD) were measured pre-operatively, and three and six months post-operatively. DMEK grafts were successfully attached and centered in all cases. No maneuver-related complications were observed intraoperatively. BCVA improved from a pre-operative 0.2 ± 0.63, to 0.43 ± 0.49 and 0.76 ± 0.51 at the three- and six-month follow-ups, respectively (*p* < 0.01). CCT decreased from a pre-operative 742 ± 118, to 546 ± 87 and 512 ± 67 at three and six months, respectively (*p* < 0.01). ECD decreased from 2878 ± 419 cells/mm^2^ to 1153 ± 466 cells/mm^2^ at three and six months, respectively (*p* < 0.01). The “Wave maneuver” may be very beneficial in DMEK cases where the AC is either very deep or the bottom of the AC is compromised. The “Wave maneuver” learning curve was brief.

## 1. Introduction

Descemet membrane endothelial keratoplasty (DMEK) is currently the first-line procedure for many corneal surgeons to treat corneal endothelial dysfunction [1]. The procedure has gained popularity since its introduction by Melles in 2002 and its first implementation on a patient in 2006 [2] due to its faster visual rehabilitation, low graft rejection rates, decreased cost, and the ability to use the anterior part of the corneal graft for anterior lamellar keratoplasty [3,4,5,6,7,8,9,10].

However, technical challenges such as careful graft preparation and meticulous graft orientation techniques demand a steep learning curve, preventing larger, more widespread DMEK adoption. Various modifications have been introduced, gradually improving surgical technique or donor preparation to succeed in challenging situations [6,7,8].

DMEK is a preferred and predictable procedure in most cases for eyes with Fuchs endothelial dystrophy (FECD) [11,12,13,14] and pseudophakic bullous keratopathy (PBK) without other ocular pathologies [15,16,17,18]. For many surgeons, several challenging ocular factors may be considered a relative contraindication for DMEK surgery [19]. The inability to flatten the anterior chamber (AC), such as in vitrectomized eyes and eyes with an intraocular lens in the anterior chamber (AC IOL), makes DMEK graft manipulation and especially the opening and centration, sometimes very difficult [20,21,22,23,24,25]. With AC IOL eyes, the graft touching the AC IOL during graft manipulation can potentially cause endothelial cell loss, become trapped in the lens haptics, or even slip into the vitreous cavity in vitrectomized eyes [23,24,25,26,27,28].

To overcome these difficulties, this maneuver is presented to reduce graft contact and manipulation over the ‘bottom’ of the AC.

## 2. Materials and Methods

### 2.1. Ethical Principles

This study followed the tenets of the Helsinki Declaration. The Hadassah Medical Center Institutional Review Board (IRB) number 0418-22HMO approval was obtained for this study, and all procedures were carried out per their guidelines.

### 2.2. Clinical Data

A retrospective review of the medical records of 27 patients who underwent a challenging DMEK surgery with the help of the “Wave maneuver” due to corneal endothelial dysfunction between 2018–2021. The surgeries took place at the Department of Ophthalmology, Hadassah Medical Center, Jerusalem, Israel, by two corneal surgeons (I.L. and E.N.N.). Surgical notes and recordings were reviewed for intraoperative complications, and the medical files were reviewed for post-operative follow-up. Post-operative follow-ups were performed at 1–3 days, one week, one month, three months, and six months after the procedure. Corneas were assessed for clearance using the slit lamp. Corneal graft attachment and central corneal thickness (CCT) were measured with Casia II anterior segment optical coherence tomography (CASIA 2, Tomey, Nagoya, Japan).

Endothelial cell count (ECC) was measured with Konan CellChek specular microscopy (Konan CellChek XL, Konan Medical, Irvine, CA, USA). Best-corrected visual acuity (BCVA) was measured with a Snellen chart.

### 2.3. Surgical Technique

Graft preparation was performed using a Sinskey hook previously described (DORC International, Zuidland, the Netherlands) [29,30], dyed with Trypan blue 0.06%, and loaded into a Geuder glass injector. Before surgery, graft size was estimated with anterior segment optical coherence tomography. The surgery began with sub-tenon anesthesia followed by a 40 mg Triamcinolone injection into the sub-tenon space, except in advanced glaucoma patients with a history of steroid responsiveness. Three limbal paracenteses of 1 mm and one main incision of 2.4 mm were created. Descemetorhexis was then made under balanced salt solution (BSS) or air maintainer in the lower-temporal paracentesis followed by graft injection into the AC, preferably in a double roll fashion facing up, and utilizing either Moutsouris “blue cannula” sign or intraoperative OCT to validate correct orientation [31]. A small air bubble was injected underneath the graft to elevate it without complete opening or centration. When the graft was faced against the posterior corneal surface in the correct orientation, it was placed in the mid periphery towards the angle to ‘lock’ it, so it will not rotate or move. Then, an air bubble was inserted under the graft, and several peripheral taps were made using the bubble bumping technique to open the graft as much as possible. Once the graft was partly open, a small air bubble was left under the graft, the AC was deepened with BSS, and with fast, swiping taps on the cornea towards the area of the desired movement, the graft was moved on the posterior corneal surface with primarily posterior stroma-DM touch (Figure 1). The AC was entirely filled with 20% SF6 or air when the graft was well centered. The patient was then taken into the ophthalmological ward in a supine position for 2–3 h and checked at the slit lamp, or with a portable slit lamp in bed in cases of vitrectomised eyes with a large iris defect or iris-intraocular lens gap. If any pupillary block or high intraocular pressure was measured, the air/gas was partially released by pressing on the lower paracentesis. Then, the patient stayed supine for at least 24 h or until gas/air disappeared from the AC (Appendix A).

A small air bubble is injected underneath the graft to elevate it. The graft is faced against the posterior corneal surface in the correct orientation, placed in the mid periphery towards the angle to ‘lock’ it so it will not move. At that point, an air bubble is placed under the graft, and several peripheral taps are made using the bubble bumping technique to open the graft as much as possible. As soon as the graft is partly open, a small air bubble is left under the graft, the anterior chamber is deepened with Balanced Salt Solution, and swiping fast taps on the cornea towards the area of the desired movement, the graft is moved on the posterior corneal surface with mostly posterior stroma Descemet membrane touch.

Patients were treated with dexamethasone 0.1%, ofloxacin, and diclofenac eye drops in the post-operative period. Ofloxacin eye drops were discontinued after five days, diclofenac eye drops were discontinued after 1–3 months (according to macular OCT), and dexamethasone eyedrops tapered down during the six months of follow-up.

### 2.4. Statistical Analysis

The statistical analysis between pre-operative and post-operative patients after three and six months with the DMEK “Wave maneuver” was performed using the Statistical Package for Social Sciences software 25.0 (SPSS Inc., Chicago, IL, USA) with two-tailed *t*-tests and with one-way analysis of variance (ANOVA) tests.

## 3. Results

During the study period, 27 eyes of 27 patients aged 22 to 90 were operated on using the DMEK “Wave maneuver” (Table 1). The ocular comorbidities are described in Figure 2. The “Wave maneuver” was successful in 25 out of the 27 cases. The graft was trapped in the angle in the two failed cases, requiring a rescue jet of BSS to be released, and conventional maneuvers were employed. All grafts were fully attached at the end of surgery with 100% air or SF6 (20%) supporting the graft. Correct orientation was confirmed at the end of each surgery. Eight eyes required a re-bubbling procedure, all performed within one month of the post-operative period, with all grafts successfully attached after the procedure. Twenty-five DMEK “Wave maneuver” eyes were evaluated three and six months after surgery. During the follow-up period, two corneas failed to clear.

Concerning visual acuity, 84% of eyes reached a BCVA of 0.50 (20/40) or better, six months after the “Wave maneuver” DMEK surgery. Increased BCVA was measured in 36%, achieving 1.0 (20/20) or better (Figure 3). The patients achieved a pinhole visual acuity of 1.36 ± 0.77 and 0.67 ± 0.51, pre-operative and six months post-operative, respectively (*p* < 0.01).

BCVA improved from a pre-operative of 0.2 ± 0.63 to 0.43 ± 0.49 and 0.76 ± 0.51 at the three and six month follow-ups, respectively (*p* < 0.01) (Figure 4).

Central corneal thickness (CCT) decreased from a pre-operative 742 ± 118 to 546 ± 87 and 512 ± 67 at three and six months (*p* < 0.01), respectively (Figure 5).

The donor endothelial cell density (ECD) decreased (from a pre-operative DMEK “Wave maneuver” operation) in measurement count of 2878 ± 419 cells/mm^2^, to 1153 ± 466 cells/mm^2^ at six months (*p* < 0.01, Figure 6).

## 4. Discussion

This series presents a novel method of performing the manipulation and placement of grafts during DMEK surgery in challenging cases. It has been called the “Wave maneuver” due to the wave-like appearance of the graft unfolding as the surgeon gently utilizes an air bubble to maneuver the graft. This maneuver aided the successful outcome of these patients, providing a statistically significant improvement in visual acuity, a statistically significant decrease in corneal thickness, and acceptable endothelial cell density at six months post-operation.

The primary complications of DMEK are partial or total graft detachment [32], which some reports rate as high as 74%, and re-bubbling rates [33]. To combat partial graft detachment in post-operative DMEK patients, gas can be injected into the anterior segment post-operatively to help approximate the graft with the remaining corneal tissue of the patient in the hope that the partial detachment will spontaneously reattach.

As with other endothelial keratoplasty procedures, using air or gas in the AC is fundamental to sutureless adherence of the donor graft to the host cornea [9,10]. Studies suggest that a larger bubble helps prevent graft detachment and rebubbling procedures, whereas gas overfill leads to complications, such as pupillary block. Pupillary block by air may lead to serious glaucomatous damage to the eye [34]. Risk factors include a previous diagnosis of glaucoma or elevated IOP, and treatment may include tapering the steroid dose [33]. Graft coverage in phakic eyes (ACD ≤ 3 mm) seems dominated by gas fill and less sensitive to patient positioning. In pseudophakic eyes with larger ACD values, the graft coverage depends on gas fill and patient positioning, the latter even more critical as ACD increases [34]. DMEK rejection is diagnosed by the presence of retrocorneal precipitates on the graft on the slit-lamp examination. A retrospective analysis of 905 eyes found a very low rejection incidence of 2.4% over four years [33].

DMEK graft unfolding often relies on a shallow and stable AC [35,36]. Due to the lack of posterior support of the vitreous, the AC in vitrectomized eyes is deep, and graft unfolding can be challenging [6,7,35,36,37]. Excessive manipulation of the donor tissue while unfolding may lead to graft failure. The injected air bubble used to tamponade the graft toward the stroma may be less effective in vitrectomized eyes due to a fluctuating iris-lens diaphragm. Injected air is apt to move posteriorly, so recurrent globe collapse is a significant problem, or the graft glides into the vitreous cavity.

Despite the many advantages of DMEK, technical difficulties in graft insertion and unfolding led to other surgical techniques. The “Wave maneuver” was developed to circumvent the “floor” problem of the anterior segment in particularly challenging eyes. While it is preferred to open the graft on the iris before elevating it onto the posterior cornea with an air bubble, the patients described here all lacked the support required due to vitrectomy (the graft could slip through the pupil posteriorly) or an AC too deep, or the risk of an AC IOL touching the endothelial cells and damaging the graft. There is a risk when moving or opening the graft in patients with synechiae. While many surgeons prefer to divert to DSAEK in these cases, the graft size is limited, and the outcome preference still lies with DMEK with the advantage of the ability to exchange almost the entire corneal endothelial layer. In this maneuver, the graft is raised at the beginning of the procedure and dragged from the Descemet’s side, not the endothelial side, avoiding cell loss.

Similarly, other surgeons have developed techniques to open or position the graft. A study described a modified double-bubble technique in DMEK for vitrectomized eyes [38]. It was the modification of a small air bubble-assisted unrolling maneuver known as the Dapena maneuver [39]. In this technique, after inserting the DMEK graft, one small air bubble was placed on top of the graft for unfolding, and another large bubble was injected beneath the graft for fixation. If peripheral edges were not attached, they applied bubble-bumping maneuvers to unfold the edges [6,7].

A slightly more complicated yet effective technique was devised whereby a corneal graft with a tail was created. A 3 mm pedicle was used to orient and drag the graft into the anterior chamber, after which the tail was extravasated. The graft was unrolled and centered using the pedicle, and gas was injected to planate the graft [40].

Another technique is called EndoGlide-DMEK [6,7]. Once an endothelium-in graft enters the AC, it unfolds easily with fewer maneuvers, but it is vital to keep the AC shallow for this technique to prevent the graft from scrolling back to the endothelium-out orientation [6,7]. EndoGlide-DMEK can assist in challenging cases such as abnormal anterior segment anatomy, gross peripheral anterior synechia, drainage devices, and filtering blebs. However, it would not be the preferred choice in the patient group described here. Rebubbling and primary graft failure rates were 11.6% and 1.5%, respectively [6,7,9,10,41]. The “Dirisamer maneuver” can similarly be utilized in cases with a shallow anterior chamber by indenting the cornea with one or two cannulas of air and then sliding over the corneal surface [29].

Most techniques described in the literature either aim to facilitate centration and movement of the graft, or help unscroll it. This maneuver achieves both objectives, and is implemented from the insertion of the graft, not after centration.

The average six-month endothelial count did decrease compared with pre-surgery, and a repeat DMEK procedure may be required at some point. Various risk factors for endothelial decompensation include, among other things, donor status, graft size and recipient factors such as glaucoma, or glaucoma surgery [42]. That being said, under the current circumstances with the multiple comorbidities, DMEK was assessed as the most appropriate procedure for these patients. It is important to note as well that repeat DMEK has been shown to be successful in cases of decompensation [43].

There are limitations to this study. A longer follow-up period would further establish the efficacy of the “Wave maneuver”. The ages and ocular histories of the patients are very heterogeneous. The stability of refractive error and possible fluctuations in astigmatism were not monitored, nor was the corneal tomography.

To conclude, presented here is a valuable technique to assist during DMEK surgeries in challenging cases to minimize graft-AC bottom contact, which aids in maintaining control over the graft and helps preserve endothelial cells.

## Figures and Tables

**Figure 1 jcm-11-05260-f001:**
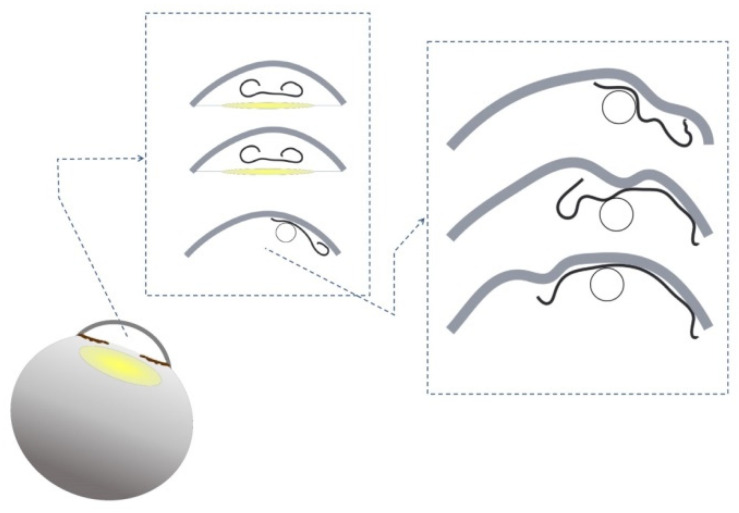
The Descemet membrane endothelial keratoplasty (DMEK) “Wave Maneuver”.

**Figure 2 jcm-11-05260-f002:**
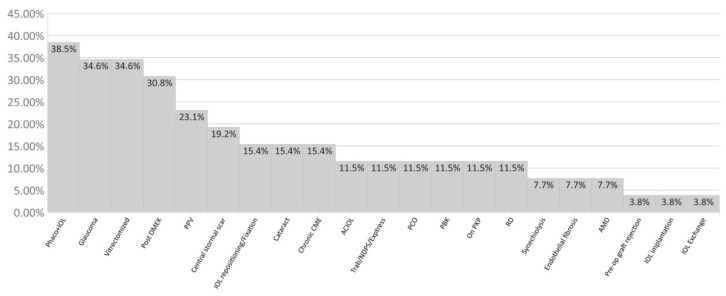
Ocular comorbidities of the patients before DMEK “Wave Maneuver”. Rehabilitation and visual acuity outcome. PPV: pars plana vitrectomy; Phaco + IOL: phacoemulsification with an intraocular lens (IOL); Pre-op: pre-operated; NPDS: non-penetrating deep sclerectomy; Elevated IOP: over 26 mm/hg corrected intraocular pressure; PCO: posterior capsule opacification; CME: cystoid macular edema; PBK: pseudophakic bullous keratopathy; PKP: penetrating keratoplasty; AMD: age-related macular degeneration; DMEK: Descemet membrane endothelial keratoplasty, RD: retinal detachment.

**Figure 3 jcm-11-05260-f003:**
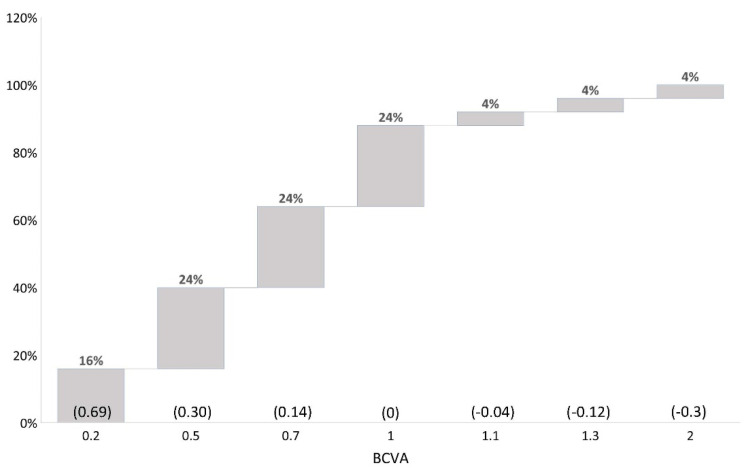
Displaying distribution of best-corrected visual acuity outcome after DMEK “Wave Maneuver” operation. LogMAR values in parentheses. BCVA: best corrected visual acuity.

**Figure 4 jcm-11-05260-f004:**
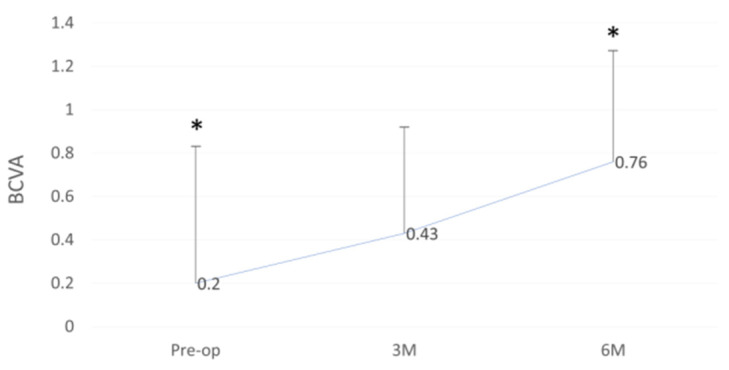
Best-corrected visual acuity (BCVA) outcome before and after DMEK “Wave Maneuver” operation. Graph displaying the BCVA before (Pre-op) and Post-op at 3 and 6 months after DMEK “Wave Maneuver” operation. BCVA: best corrected visual acuity. M: months. Asterisk represents statistical significance (*p* < 0.01) for Pre-op versus 6M.

**Figure 5 jcm-11-05260-f005:**
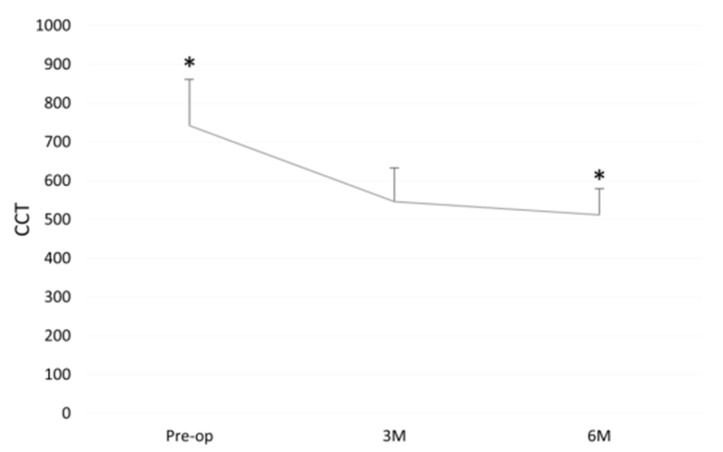
Central corneal thickness (CCT) before and after DMEK “Wave Maneuver” operation. Graph displaying the CCT before (Pre-op) and Post-op at 3 and 6 months after DMEK “Wave Maneuver” operation. CCT: central corneal thickness. M: months. Asterisk represents statistical significance (*p* < 0.01) for Pre-op versus 6M. Donor endothelial cell density (ECD).

**Figure 6 jcm-11-05260-f006:**
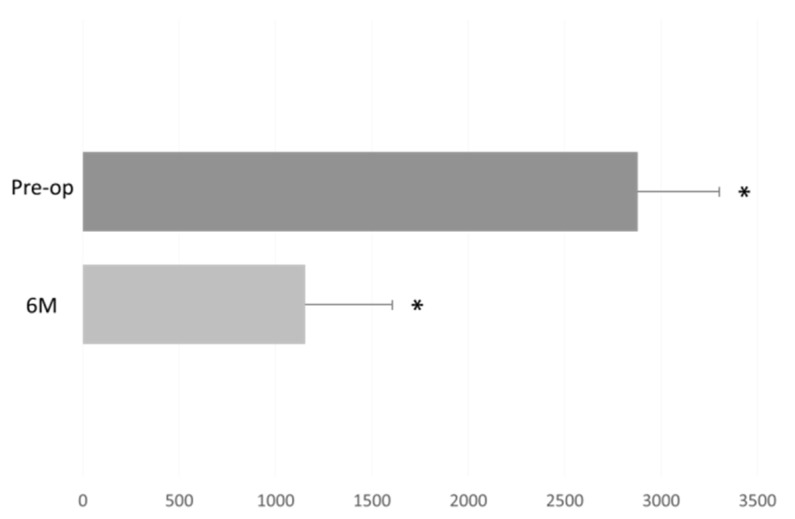
Donor endothelial cell density (ECD) before and after DMEK “Wave Maneuver” operation. Graph displaying the donor ECD before (Pre-op) and six months after DMEK “Wave Maneuver” operation. M: months. Asterisk represents statistical significance (*p* < 0.01) for Pre-op versus 6M.

**Table 1 jcm-11-05260-t001:** Demographics for patients who underwent DMEK “Wave Maneuver”.

	Number	Percentage
Patients (n)	27	
Eyes/eyes consecutive	27/25	92.6%
Age (mean and S.D.)	67 ± 21	
Men	14	51.85%
Women	13	48.15%

## Data Availability

Hadassah Medical Center, Ein Kerem. P.O. Box 12271, Jerusalem, 9112102 Israel.

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
