# Peer review of "The Descemet Membrane Endothelial Keratoplasty (DMEK) “Wave Maneuver”"

_jcm, 2022, doi:10.3390/jcm11185260_

Round 1
Reviewer 1 Report
Thank you for your well written report describing a modification on unscrolling and centring DMEKs in complex eyes. The video and technique are very well described and the case series result impressive. I have outline a few grammatical corrections to consider amending at the end. Two issues I would hope the authors would address either in the introduction or conclusion:
1. How does the wave maneuver in terms of centrating differ from using an airbubble and corneal sweeps to centre a DSAEK. I include reports from the DSAEK literature that may be useful to reference and include a paragraph how these maneuvers have evolved.
Mark Terry Ophthalmology 115(7) 2008 1179-1186 where they describe “If the donor disc is off from centration by ≤2 mm, gentle sweeping and compression of the surface of the cornea across the entire length of the donor disc diameter ballots and moves the tissue in the direction of the sweep until the desired position is obtained.” Also the differences with the use of a “Cindy Sweepr in the same paper: With the chamber completely filled with air, the specialized “Cindy Sweeper” (Bausch and Lomb, St. Louis, MO) is used to compress the surface of the cornea and sweep from the center to the periphery, repeatedly for about 2 minutes. This maneuver “milks” any interface fluid out to the edge of the graft and into the anterior chamber, thereby stabilizing the graft. Sweeping of the surface is performed until the surgeon feels that the edges of the graft remain perfectly aligned with the overlying template mark and without any movement of the graft at all
2. The Endothelial cell loss quoted is on the higher side compared to the literature. I include 3 references below. I do feel the authors need to justify this as the data suggests these DMEKs will not last at least 10years and is this an issue with the technique itself, the complicated ocular pathologies or donor factors. Please address
Endothelial Cell Loss After Descemet's Membrane Endothelial Keratoplasty for Fuchs' Endothelial Dystrophy: DMEK Compared to Triple DMEK,
American Journal of Ophthalmology,
Volume 218,
2020,
Pages 1-6,
ISSN 0002-9394,
https://doi.org/10.1016/j.ajo.2020.05.003.
The mean preoperative donor ECDs were 2,630 ± 194 cells/cm2 and 2,643 ± 197 cells/cm2 in DMEK-only and triple-DMEK groups, respectively. At 1 month, the mean ECDs were 1,968 ± 476 cells/cm2 and 1,737 ± 422 cells/cm2 in DMEK-only and triple-DMEK groups, respectively, representing ECL of 25% and 35%, respectively, from preoperative donor ECDs. At 1 year, mean ECDs were 1,748 ± 427 cells/cm2 and 1,511 ± 437 cells/cm2, respectively, in DMEK-only and triple-DMEK groups, representing ECL of 33% and 41%, respectively.
Winston Chamberlain, Charles C. Lin, Ariana Austin, Nicholas Schubach, Jameson Clover, Stephen D. McLeod, Travis C. Porco, Thomas M. Lietman, Jennifer Rose-Nussbaumer,
Descemet Endothelial Thickness Comparison Trial: A Randomized Trial Comparing Ultrathin Descemet Stripping Automated Endothelial Keratoplasty with Descemet Membrane Endothelial Keratoplasty,
Ophthalmology,
Volume 126, Issue 1,
2019,
Pages 19-26,
ISSN 0161-6420,
https://doi.org/10.1016/j.ophtha.2018.05.019.
(https://www.sciencedirect.com/science/article/pii/S0161642018310339)
Abstract: Purpose
To compare clinical outcomes of ultrathin Descemet stripping automated endothelial keratoplasty (UT-DSAEK) and Descemet membrane endothelial keratoplasty (DMEK) in the treatment of corneal endothelial dysfunction.
Design
Patient and outcome-masked, randomized controlled clinical trial.
Participants
Patients with damaged or diseased endothelium from Fuchs endothelial dystrophy or pseudophakic bullous keratopathy who were considered good candidates for DMEK or UT-DSAEK.
Methods
Study eyes were randomized by the eye bank to UT-DSAEK or DMEK 1 to 2 days before surgery.
Main Outcome Measures
The primary outcome of the trial was best spectacle-corrected visual acuity (BSCVA) at 6 months. Secondary outcomes included 3- and 12-month BSCVA; 3-, 6-, and 12-month endothelial cell counts; intraoperative and postoperative complications; and change in pachymetry.
Results
A total of 216 patients with endothelial dysfunction were screened, and 50 eyes of 38 patients were enrolled by 2 surgeons at Casey Eye Institute at Oregon Health & Science University in Portland, Oregon, and at Byers Eye Institute at Stanford University in Palo Alto, California. Overall, we found DMEK to have better visual acuity outcomes compared with UT-DSAEK after correcting for baseline visual acuity: compared with UT-DSAEK, those randomized to DMEK had 1.5 lines better BSCVA at 3 months (95% confidence interval [CI], 2.5–0.6 lines better; P = 0.002), 1.8 lines better BSCVA at 6 months (95% CI, 2.8–1.0 lines better; P < 0.001), and 1.4 lines better BSCVA at 12 months (95% CI, 2.2–0.7 lines better; P < 0.001). Average endothelial cell counts were 1963 cells/mm2 in DMEK and 2113 cells/mm2 in UT-DSAEK at 6 months (P = 0.17) and 1855 cells/mm2 in DMEK and 2070 cells/mm2 in UT-DSAEK at 12 months (P = 0.051). Intraoperative and postoperative complication rates were similar between groups.
Conclusions
Descemet membrane endothelial keratoplasty had superior visual acuity results compared with UT-DSAEK at 3, 6, and 12 months in patients with isolated endothelial dysfunction with similar complication rates.
3. Did you time how long the maneuver took to unscrol the DMEK. Please give some idea on average times and compare to the literture conclusion below:
Sáles, Christopher S. MD, MPH; Terry, Mark A. MD; Veldman, Peter B. MD; Mayko, Zachary M. MS; Straiko, Michael D. MD. Relationship Between Tissue Unscrolling Time and Endothelial Cell Loss. Cornea: April 2016 - Volume 35 - Issue 4 - p 471-476
doi: 10.1097/ICO.0000000000000771
Once the DMEK tissue is safely in the anterior chamber, surgeons need not rush the “DMEK dance” because longer unscrolling times may not endanger the endothelium.
Grammar:
Pg 1 line 9 replace inferior with posterior
Line 10 (ac)
Line 11 (ACIOLs)
Line 12 27 eye with complex ocular pathology.
Line 31 -omit supurb
Line 39 start sentence: DMEK is a preferred and predictable procedure in most case of eyes with….
Line 41 (15-18). For many surgeons, eyes with concurrent ocular pathologies such as ACIOL, glaucoma shunts or being vitrectomised may be ….
Page 5 Fig 2 ?enteral stromal scar . Is this CENTRAL?
Also my preference would be to have actual patient numbers 1-9 rather than percentages
Thank you for this robust case series and excellent descriptiona nd honest presentation of the results.
Author Response
The point by point responses are in the attachement

Reviewer 2 Report
Very interesting article, congratulations to the authors, very nice videos too. It is indeed challenging to perform DMEK in abnormal ACs and every effort is welcome.
Just some minor comments:
- Regarding the results and Figure 3, I would also mention the conversion of visual acuities from decimal to LogMar.
- Line 140, "respectively" should be out of the brackets.
- Maybe go through figures 2 and 3 to make them more visual?
- I would "tidy up" a little bit the discussion, for instance in lines 196-198 DMEK rejection is discussed after pupillary block and then graft coverage is mentioned, which is more related to pupillary block.
- I would compare the results of VA with the current literature.
Author Response
The point by point responses are in the attachment
